# How do tradeoffs in satellite spatial and temporal resolution impact snow water equivalent reconstruction?

Edward H. Bair[1], Jeff Dozier[2], Karl Rittger[3], Timbo Stillinger[1], William Kleiber[4], and Robert E. Davis[5]

[1]Earth Research Institute, University of California, Santa Barbara, CA USA 93106
[2]Bren School of Environmental Science and Management, University of California, Santa Barbara, CA USA 93106
[3]Institute of Arctic and Alpine Research, University of Colorado, Boulder, CO 80309
[4]Department of Applied Mathematics, University of Colorado, Boulder, CO 80309
[5]Cold Regions Research and Engineering Laboratory, Hanover, NH USA 03755

*Correspondence to*: Edward H. Bair (nbair@eri.ucsb.edu)

**Abstract**

Given the tradeoffs between spatial and temporal resolution, questions about resolution optimality are fundamental to the study of global snow. Answers to these questions will inform future scientific priorities and mission specifications. Heterogeneity of mountain snowpacks drives a need for daily snow cover mapping at the slope scale ($\leq$ 30 m) that is unmet for a variety of scientific users, ranging from hydrologists to the military to wildlife biologists. But finer spatial resolution usually requires coarser temporal or spectral resolution. Thus, no single sensor can meet all these needs. Recently, constellations of satellites and fusion techniques have made noteworthy progress. The efficacy of two such recent advances is examined: 1) a fused MODIS - Landsat product with daily 30 m spatial resolution; and 2) a harmonized Landsat 8 - Sentinel 2A/B (HLS) product with 3-4 day temporal and 30 m spatial resolution. State-of-art spectral unmixing techniques are applied to surface reflectance products from 1 & 2 to create snow cover and albedo maps. Then an energy balance model was run to reconstruct snow water equivalent (SWE). For validation, lidar-based Airborne Snow Observatory SWE estimates were used. Results show that reconstructed SWE forced with 30 m resolution snow cover has lower bias, a measure of basin-wide accuracy, than the baseline case using MODIS (463 m cell size), but greater mean absolute error, a measure of per-pixel accuracy. However, the differences in errors may be within uncertainties from scaling artifacts e.g., basin boundary delineation. Other explanations are 1) the importance of daily acquisitions and 2) the limitations of downscaled forcings for reconstruction. Conclusions are: 1) spectrally unmixed snow cover and snow albedo from MODIS continue to provide accurate forcings for snow models; and 2) finer spatial and temporal resolution through sensor design, fusion techniques, and satellite constellations are the future for Earth observations, but existing moderate resolution sensors still offer value.

## 1. Introduction

Mountain snowpacks are challenging for remote sensing because they change rapidly. Moderate resolution sensors such as MODIS and VIIRS image Earth daily, but at resolutions (463 m - 750 m) that cannot resolve slope scale features of interest to a variety of scientific users ranging from hydrologists (Blöschl, 1999), to the military (Vuyovich et al., 2018), to wildlife biologists (Conner et al., 2018). Finer resolution multispectral sensors such as Landsat 8/9 provide spatial resolutions of 30 m, but at 16-day revisits, during which time the snow cover can change considerably. Because of cloud cover, useable optical imagery with such infrequent revisits can be months apart. Recognizing that no single satellite/instrument can provide fine spatial and temporal resolution, constellations of satellites with coordinated overpass times have emerged. Two examples are the Sentinel 2 A/B and Landsat 8/9 pairs. For optical bands, the Sentinels image Earth every 5 days at 20 m, and Landsats 8/9 image Earth every 8 days at 30 m. The Harmonized Landsat 8/9 Sentinel 2 (HLS) product (Claverie et al., 2018) improves the average revisit time to 3-4 days at 30 m spatial resolution.

Effects of snow cover estimates at finer resolution have been examined in a few studies, showing a wide range of improvements in errors. In comparing snow cover depletion curves from Landsat MSS (80 m pixels; 16-day repeat) and AVHRR imagery (1100 m pixels; daily repeat), Baumgartner et al. (1987) found that AVHRR tended to overestimate snow cover where it was patchier (lower elevations) and underestimate snow cover where it was more widespread (higher elevations), relative to MSS. They concluded that AVHRR imagery could be used to fill in temporal gaps in depletion curves generated from Landsat MSS. Luce et al. (1998) compared a spatially explicit SWE model at 30 m with single and two point models for a small basin in Idaho. The 30 m model showed significantly lower errors than the single and two point models. Cline et al. (1998) examined the effect of upscaling the spatial resolution of a DEM and snow cover in an energy balance SWE model at a range of resolutions: 30, 90, 250 and 500 m. Positive biases in the coarser resolution estimates arising solely from basin delineation artifacts were reported, thus the authors advise using vector basin outlines (as was done in Section 2.5). When these artifacts were corrected, the SWE volumes at 90 m were overestimates while those at coarser resolutions were underestimates. Blöschl (1999) examined scaling issues in snow hydrology and shows that pixel sizes of a few m are needed to accurately capture basin-scale SWE. Turpin et al. (2000) examined snow cover maps derived from AVHRR and Landsat TM (30 m resolution; 16-day repeat) and report discrepancies, also finding that AVHRR failed to resolve patchy snow compared to TM. Durand et al. (2008) were the first to create a fused MODIS and Landsat product. For the coarse resolution product they used binary snow cover from MODIS (Hall et al., 2002). For the fine resolution product they used Landsat 7 ETM+ surface reflectance in a spectral unmixing algorithm (Painter et al., 2003). The authors then used a linear program, constrained to match the ETM+ fractional snow-covered area (fsca) imagery while also matching the daily changes in fsca observed by MODIS. Applying their linear program to the upper Rio Grande, the authors found differences in fsca between the ETM+ fsca, the MODIS fsca, and the fused product. When run through a snow reconstruction model, these differences equated to a 51% reduction in mean absolute error (MAE) and a 49% reduction in bias for SWE using the fused snow cover versus the ETM+ snow cover. Using the same reconstruction model, Molotch and Margulis (2008) report a 23% MAE in SWE

using ETM+ snow cover, versus a 50% MAE in SWE using MODIS snow cover, and 89% MAE using AVHRR snow cover. Rittger et al. (2013) examined spectrally unmixed snow cover from ETM+ (similar to the approach in Painter et al., 2009) and several approaches for mapping snow cover from MODIS, including spectral mixture analysis (Painter et al., 2009). They found that ETM+ mapped consistently more patchy snow cover than the MODIS approaches, suggesting fewer false negatives and thus a greater recall statistic. Winstral et al. (2014) examined scale in a snow energy balance model at a range of spatial resolutions and find that 100 m spatial resolution is needed to accurately simulate snow melt. Contrary to Cline et al. (1998), Schlögl et al. (2016) report that SWE increases with DEM resolution in two alpine basins. Similarly, Baba et al. (2019) used an energy balance model with a DEM at 8-1000 m and report good agreement with fine resolution snow cover maps up to 250 m, but a loss in agreement at coarser resolution, likely due to excessive smoothing of topographic effects. Rittger et al. (2021) used a random forest to fuse spectrally unmixed snow cover from MODIS with Landsat 5 and 7 ETM+. The authors' comparisons show sharper snowlines (transition from no snow to fully snow covered) in the Landsat and fused imagery compared to MODIS, again indicating that Landsat may have greater recall than MODIS in this difficult to validate region. Bouamri et al. (2021) examined differences between snowmelt models with and without solar radiation represented in the Atlas Mountains of Morocco. Although the models with solar radiation better simulated the snowcover used for validation, aggregating the simulated snow cover from 100 to 500 m suppressed those improvements. In summary, many studies have compared coarse and fine resolution snow cover, but only three studies to our knowledge (Cline et al., 1998; Durand et al., 2008; Molotch and Margulis, 2008) have examined the impact of resolution on SWE reconstruction, all finding significant improvements from finer spatial resolution. Since those studies, considerable advances have been made in SWE reconstruction techniques (Bair et al., 2016; Rittger et al., 2016) as well as snow cover (Stillinger et al., 2023) and albedo mapping (Bair et al., 2019), hence the justification for revisiting the effects of spatial and temporal resolution.

## 2.   Approach

Three daily snow cover estimates were used to force a SWE reconstruction model at two spatial resolutions: a baseline at 463 m using MODIS with the Snow Property Inversion from Remote Sensing (SPIReS, Section 2.1, Bair et al., 2021), and two 30 m estimates, one from the Harmonized Landsat-Sentinel (HLS) surface reflectance product (also using SPIReS, Section 2.2), the other from Snow Covered Area and Grain Size (SCAG)-Fusion (Section 2.3). The period covered is 1 Jan 2018 to 31 Dec 2020, limited by the intersection of the availability of the SCAG-Fusion and the HLS. The domain is the Tuolumne River Basin above the Hetch Hetchy Reservoir in the Sierra Nevada USA because of the availability of Airborne Snow Observatory (Painter et al., 2016) estimates of SWE for validation. This approach rests on the hypothesis that 1) fsca and 2) snow albedo are the two most important variables in SWE reconstruction (see Section 2.4). The importance of fsca in SWE reconstructions can be traced to several studies (e.g., Durand et al., 2008; Molotch and Margulis, 2008). The importance of snow albedo in SWE reconstructions is shown in Bair et al. (2019).

## 2.1. SPIReS-MODIS

The baseline case uses SPIReS to map snow cover from MODIS at 463 m daily resolution, although the effective pixel size can be up to 5× as large for off nadir acquisitions (Wolfe et al., 1998; Dozier et al., 2008). The MOD09GA daily surface reflectances (Vermote and Wolfe, 2015) are unmixed into fsca and properties used to model albedo (grain size and dust concentration). These estimates are then run through a series of filters including persistence filters for clouds and time-based smoothing/interpolation.

## 2.2. SPIReS-HLS

     One daily 30 m snow cover product used also comes from the SPIReS approach applied to the HLS. As the first snow mapping application, to our knowledge, of HLS data. Thus, we describe the workflow in more detail than SPIReS-MODIS. The Tuolumne River Basin above Hetch Hetchy Reservoir straddles two Sentinel tiles, so HLS version 1.4 multi-band HDF files from both tiles were downloaded (https://hls.gsfc.nasa.gov/). For calendar years 2018-2020, four combinations of

products were downloaded: two tiles (11SKB,11SKC); and two products – S30 (Harmonized Sentinel-2 MSI) and L30 (Harmonized Landsat-8 OLI). We attempted to download the newer HLS version 2.0 from NASA Earthdata Search, but as of this writing, the S30 product for those tiles only extends back to 2020 Sep 23 because of daily limits on the number of Sentinel-2A/B scenes that can be downloaded by NASA from ESA for reprocessing. Seven bands covering visible through shortwave-infrared wavelengths were used from each sensor: 1-4, 8A, and 11-12 for S30; 1-7 for L30. Mean local solar geometry was

obtained from the accompanying header files. The multi-band images were stacked, mosaiced, and cropped to the basin to form a $1119 \times 1297 \times 7 \times 311$ 4D data structure, with dimensions of rows, columns, bands, and time. Each Landsat has a 16-day revisit, thereby providing imagery at 8-day intervals for each tile, and each Sentinel has a 10-day revisit, providing imagery at 5-day intervals. Thus, combined revisits ranged from around 1 day to 10 days with a mean of 3.5 days (Figure 1). Theoretical revisit times estimated before Landsat 9 was launched (Li and Roy, 2017) for a three-satellite constellation at this basin's

latitude shows a mean of 3.8 days, with a minimum of less than one day and a maximum of 7.0 days. There are only 4 revisit times greater than 7 days shown in Figure 1; all other observations lie within the theoretical revisit times.

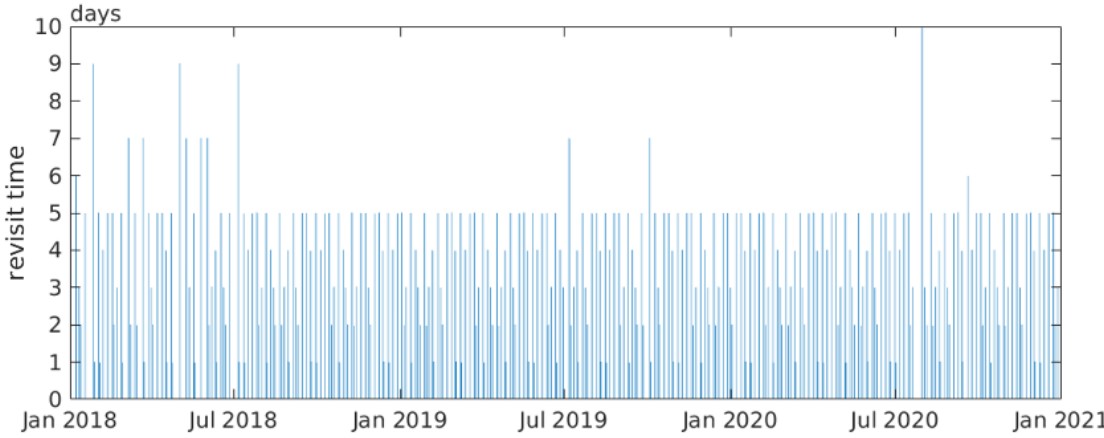

Figure 1:

*Revisit times covering the Tuolumne River Basin above Hetch Hetchy Reservoir for the HLS S30 and L30 combined.*

Red/green/blue band imagery for each day was examined visually. Days with clouds or incomplete spatial coverage over the watershed (many images have large areas with no data) were discarded. After filtering, 156 or about half of the days were kept. The minimum, median, and maximum time spacings between acquisitions after filtering were 1, 5, and 40 days. The SPIReS spectral unmixing approach was then applied to these filtered surface reflectances as described for Landsat 8 OLI in Bair et al. (2021), yielding the variables fsca, snow grain size, and dust concentration. A per-pixel spline interpolation was
applied to each of the variables in the time dimension to make them continuous, covering all days from 2018-2020.

### 2.3. SCAG-Fusion

A second daily 30 m snow cover product used was a MODIS-Landsat fusion, created using two random forests for classification and regression based on previous work (Rittger et al., 2021) but retrained using Landsat 8 OLI data. Standard cloud masks (Foga et al., 2017) were used to select the 100 most cloud free Level 2 surface reflectance images (USGS, 2021)
for dates spanning Mar 2013 to Mar 2021 (Figure 2). Winter months had fewer training data than summer months. Six scenes were manually removed after visual inspection of red/green/blue imagery. Because the initial filtering did not remove all clouds, a second cloud filtering step with Superpixels and Gabor filtering was used (Stillinger, 2019). This second step removed all the clouds, but removed some snow cover as well. These filtered Landsat 8 surface reflectances along with MODIS MOD09GA surface reflectances were unmixed into fsca and snow surface properties that affect albedo (Painter et al., 2009;
Painter et al., 2012). This SCAG spectral unmixing differs from the SPIReS approach; it finds the best fit from an endmember library for the snow-free parts of the pixel, whereas SPIReS uses an empirical snow-free endmember. There are other differences in the treatment of light absorbing impurities, filtering, and time-space smoothing (Rittger et al., 2020). For more details and a recent comparison between SPIReS, SCAG, and all other accessible snow mapping algorithms see Stillinger et al. (2023). Estimates of fsca and snow surface properties that affect albedo i.e., grain size and visible albedo degradation were

used as training data. Physiographic variables, including solar illumination and land classification were used as predictors. The two-step approach consists of an initial model that classifies pixels into 3 cases: (1) 0%, (2) 100%, or (3) 1-99% fsca. For case (3), a second regression random forest was used to estimate fsca on the 1-99% interval. This two-step classification-regression approach was found to be less biased at predicting 100% snow covered pixels than using a single-step random forest predicting 0-100% fsca.

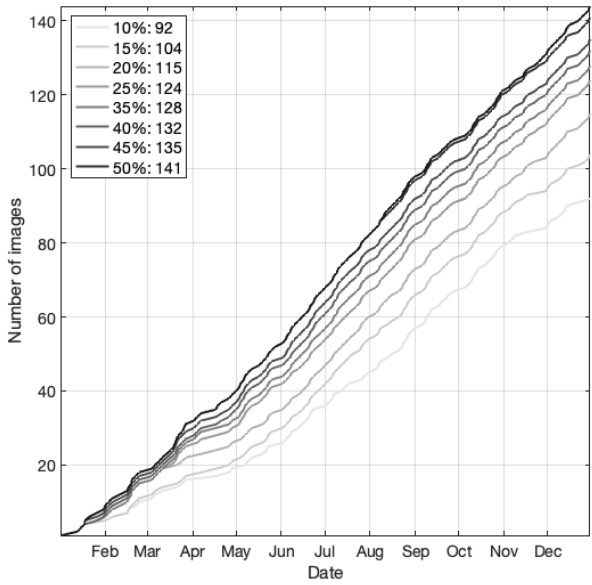


Figure 2:

*Cumulative number of images over 8 years acquired from Landsat 8. Lighter to darker lines indicate increasing cloud coverage from 10-50%. Numbers after the cloud cover percentage in the legend correspond to the total number of images. Images with 12% or less cloud cover were selected.*

### 2.4. Parallel energy balance

150         At an hourly timestep, the Parallel Energy Balance model ParBal (Bair et al., 2016; Rittger et al., 2016; Bair et al., 2018) downscales state and flux variables solving for the surface snow energy balance. The computed melt is multiplied by the fsca and summed backward from end-of-melt to peak SWE for each pixel to estimate SWE on the ground throughout the melt season. Evaluations of ParBal (Bair et al., 2016; Bair et al., 2018) forced with snow cover from the MODSCAG approach

(Painter et al., 2009; Rittger et al., 2020) show a mean absolute error (MAE) of 22-26% using SWE from ASO for validation. There are two significant changes to ParBal here. 1) U and V wind component forcings use the hourly MERRA-2 (Global Modeling and Assimilation Office (GMAO), 2015) data instead of N/GLDAS-2 (Rodell et al., 2004; Xia et al., 2012). Using U & V components with global forcings allows for terrain-based wind downscaling using curvature and slope (Liston et al., 2007), whereas GLDAS only provides wind speed. 2) A new estimate of SWE on the ground, called hybrid SWE, leverages

GLDAS SWE (the GLDAS NOAH 3-hour 0.25° v2.1 model was used) and captures the accumulation phase. Previously,

ParBal estimates were limited to the ablation phase only. The concept is to identify GLDAS pixels with similar snow cover duration as the fine-scale fsca pixels, find the peak SWE day from those GLDAS pixels, then scale the GLDAS estimates by the ParBal SWE estimate on that peak day. This process is repeated for every fine-scale pixel. The GLDAS SWE, $SWE_{GLDAS}$, is extracted for the domain, in this case a bounding box covering the Tuolumne River Basin above Hetch Hetchy Reservoir.

Pixels with the same snow cover duration are identified by the logical vector $t$ as

$$t = \left(SWE_{GLDAS,\Delta t_1} > 0, fsca_{\Delta t_1} > 0\right) \& \left(SWE_{GLDAS,\Delta t_2} = 0, fsca_{\Delta t_2} = 0\right)$$
$$SWE^*_{GLDAS} = SWE_{GLDAS,t} \tag{1}$$

where the asterisk denotes the selected pixels and fsca is from the fine-scaled product i.e., Section 2.1 - 2.2. The $\Delta t_1$ and $\Delta t_2$ indicate different time periods (days). Because there can be multiple pixels with matching snow cover duration, the daily mean of $SWE^*_{GLDAS}$ is taken. The maximum value of that daily mean and its index $imax$ are computed. A scaling coefficient $c$ is calculated as

$$c = SWE_{ParBal,imax}/max(\overline{SWE^*_{GLDAS}}) \tag{2}$$

where $SWE_{ParBal,imax}$ is the value of the reconstructed SWE from ParBal (Eq. 2 in Bair et al., 2016) at the time $imax$ and the overbar denotes an average. The following case can arise

$$c = 0, SWE_{ParBal,imax} = 0, (SWE_{GLDAS} > 0) \& (fsca > 0). \tag{3}$$

For example, this case can occur when ParBal models all the mass loss via sublimation. For this case, $SWE_{GLDAS}$ is used when fsca $> 0$. Otherwise, the hybrid SWE prior to the peak is set at day $i$ as

$$SWE_{hybrid,i} = c \times SWE_{GLDAS,i} , i \leq imax. \tag{4}$$

This scaling can cause unrealistic daily increases and decreases in SWE; thus a smoothing spline is applied. This hybrid

SWE has yet to be evaluated throughout the accumulation season, but comparison with the reconstructed SWE during the ASO acquisitions show negligible differences, indicating at least the $imax$ estimate is occurring roughly at the right time of year since the all of the ASO flights examined here took place during the ablation season (with the exception of 13 Apr 2020, Section 3). Figure 3 shows this hybrid GLDAS and reconstructed SWE for an example pixel in WY 2019. ParBal was run with each of the snow cover forcings, holding all other inputs constant.

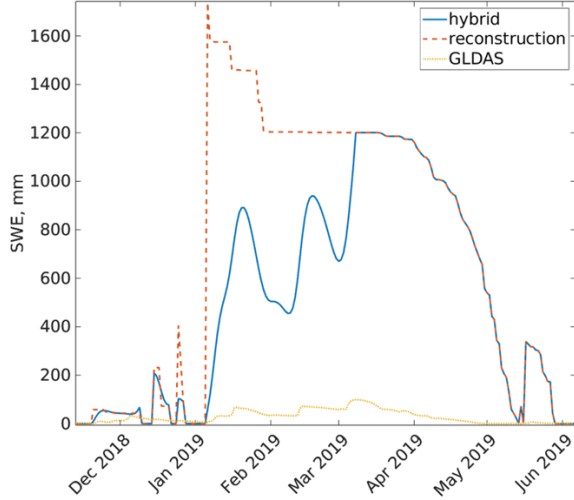


Figure 3:

*Hybrid SWE estimates for the accumulation season combining reconstruction and GLDAS for an example pixel using the SPIReS-HLS snow cover.*

### 2.5. Airborne Snow Observatory

ASO 50 m SWE estimates for the Tuolumne River Basin above the Hetch Hetchy Reservoir, which is the most sampled basin by ASO, were downloaded from the National Snow and Ice Data Center for 2018 and 2019 and from ASO Inc. for 2020 (Table 1). The number of acquisitions per year ranged from two (2018) to four (2019) with a total of nine. Accuracy of ASO measurements at the basin scale cannot be estimated directly from data, since there is no better method for validation, but since 2021, ASO has provided basin-wide uncertainty estimates on their reports available on their website

(https://www.airbornesnowobservatories.com), mostly based on uncertainty in modeled density, with a small uncertainty in depth. The reported mean basin-wide uncertainty in SWE for ASO flights the entire Tuolumne River Basin for 2021 and 2022 is ±4%, so we assume similar errors in 2018-2020 and use that uncertainty estimate.

| Year | Name | Mean SWE, mm |
|---|---|---|
| 2018 | 23 Apr | 418 |
|  | 28 May | 127 |
| 2019 | 17 Apr | 1095 |
|  | 3 May | 840 |
|  | 13 Jun | 441 |
|  | 5 Jul | 111 |
| 2020 | 13 Apr | 293 |
|  | 7 May | 191 |

| | 21 May | 128 |
|---|---|---|

Table 1:

*Tuolumne River Basin above Hetch Hetchy Reservoir SWE estimates for 2018-2020 for the Airborne Snow Observatory.*

## 2.6. Analysis

The ASO images were resampled from a cell size of 50 m to 2000 m (4× the MODIS resolution) and 120 m (4× the Landsat resolution), using a mean-preserving technique with a weighted resampling covering the image (mapresize, MathWorks, 2022). The ASO images were kept in their native UTM 11N projection. The upscaled cell sizes account for geolocational and sensor-to-sensor uncertainty of 1-2 pixels for MODIS and Landsat/Sentinel-2 (Tan et al., 2006; Storey et al., 2016). The ASO dates in Table 1 were extracted for each of the three SWE reconstructions. Then, the matched baseline SPIReS-MODIS images were upscaled from 463 m to 2000 m and reprojected from a sinusoidal projection to UTM 11N. The matched MODIS-Landsat fusion and SPIReS-HLS images were upscaled from 30 m to 120 m but kept in their native UTM 11N projection. Vectors of the Tuolumne River Basin above Hetch Hetchy Reservoir were obtained from ASO Inc. These vectors were then converted into coarse resolution masks of the basin. Water bodies and other areas with no data in either the ASO images or in the SWE reconstructions were removed from the masks. Areas outside of the masks were then set to null values. These common masks were applied to the upscaled SWE reconstruction and ASO images such that areas outside the masks were set to null values. The upscaled ASO images were compared with the upscaled SWE reconstruction images. The following error statistics were computed for a given date: bias as a measure of basin-wide error, relative bias normalized by ASO mean SWE, mean absolute error (MAE) as an unweighted measure of per-pixel error, and relative MAE normalized by ASO mean SWE:

$$\text{bias} \ = \ \frac{1}{N} \sum_{j=1}^{N} \text{SWE}_{ParBal,j} \ - \text{SWE}_{ASO,j} \tag{5}$$

$$\text{relative bias} \ = \ \frac{\frac{1}{N} \sum_{j=1}^{N} \text{SWE}_{ParBal,j} \ - \text{SWE}_{ASO,j}}{\frac{1}{N} \sum_{j=1}^{N} \text{SWE}_{ASO,j}} \tag{6}$$

$$\text{mean absolute error} \ = \ \frac{1}{N} \sum_{j=1}^{N} \left| \text{SWE}_{ParBal,j} \ - \text{SWE}_{ASO,j} \right| \tag{7}$$

$$\text{relative mean absolute error} \ = \ \frac{\frac{1}{N} \sum_{j=1}^{N} \left| \text{SWE}_{ParBal,j} \ - \text{SWE}_{ASO,j} \right|}{\frac{1}{N} \sum_{j=1}^{N} \text{SWE}_{ASO,j}} \tag{8}$$

where $N$ the total number of pixels and $j$ is an individual pixel. Mean values of the four error statistics were also averaged by year. MAE is used instead of Root Mean Squared Error because it evenly weights errors which is preferred when comparing modeled values (Willmott and Matsuura, 2005) i.e., ParBal to ASO, neither of which directly measure SWE.

## 2.7. Snow albedo errors

Errors in snow albedo directly impact the accuracy of reconstructed SWE (Bair et al., 2019). However, for the spectral unmixing approaches used here, the albedo errors are low, evaluated using terrain-corrected measurements from Mammoth Mountain (e.g., Bair et al., 2022), only 23 km from Mount Lyell, the highest point in the Tuolumne River Basin. For example, from water years 2017-2019, the Root Mean Squared Error (RMSE) for MODIS-SPIReS, calculated using the best value for a 3×3 neighborhood around the validation site, is 2.3% with no bias (Table 2). These albedo errors are similar to the accuracy of the HDRF surface reflectance products, evaluated over dark targets (Vermote et al., 2016; Bair et al., 2022). These improvements in remotely sensed snow albedo over previous assessments, showing RMSE values of 4.6 to 4.8% with 0.7-1.3% bias for MODIS (Bair et al., 2019; Bair et al., 2021), come from improved cloud snow discrimination filters and adjustments to thresholds such as the minimum grain size for dirty snow (Section III-J of Bair et al., 2021).

| Water year | Bias, % | RMSE, % |
|---|---|---|
| 2017 | -0.8 | 2.2 |
| 2018 | -0.3 | 2.4 |
| 2019 | 1.0 | 2.4 |
| **mean** | **0.0** | **2.3** |

Table 2:

*Snow albedo from SPIReS-MODIS validated with terrain-corrected snow albedo from the CUES site on Mammoth Mountain (Bair et al., 2015) taken using an adjustable arm to keep the radiometers 1 m above the snow surface (Bair et al., 2022). A best of 3×3 pixel neighborhood was used to account for geolocational uncertainty.*

We are not dismissing errors in albedo, as these remotely-sensed snow albedo errors can lead to 5-11 % MAE for reconstructed SWE (Bair et al., 2019), but without independent measurements of spatially distributed albedo, we lack validation data for further error evaluation.

## 3. Results and discussion

Basin-wide mean SWE errors are shown in Figure 5a-c and in Table 3. The lower mean MAE, by 10-14% (bold in Table 3) from SPIReS-MODIS is perhaps the most intriguing result, contradicting the result of previous studies (Section 1) which find that finer spatial resolution estimates of snow cover reduce SWE errors. The reduction of 4-5% relative bias from the two 30 m snow cover forcings compared to MODIS agrees with the previous findings, although the magnitudes of the reductions are smaller than in previous studies (e.g., Durand et al., 2008). To test if the lower MAE from MODIS are resolution artifacts, the SPIReS-HLS and SCAG-Fusion products were also upscaled to 2000 m cell sizes instead of 120 m. For mean values over all water years for these upscaled comparisons (Table A1), SPIReS-MODIS still had the lowest relative MAE, but the SCAG-Fusion relative bias dropped to 3% while the SPIReS-HLS relative bias increased to 9%, equal to SPIReS-MODIS. These results suggest that the evaluations are sensitive to the upscaled pixel size, meaning that the differences in errors across the

three SWE reconstructions may be within uncertainty bounds introduced by upscaling artifacts such as basin delineation. For example, in the UTM 11N projection, the shapefile obtained from ASO Inc. for the Tuolumne River Basin above Hetch Hetchy

Reservoir has an area of 1175 km$^2$; a raster of the basin at 120 m has an area of 1153 km$^2$ (-1.8%) while a raster at 2000 m has an area of 1132 km$^2$ (-3.6%). Even when using vector basin outlines, as suggested by Cline et al. (1998), these artifacts are inherent in the discretization of geospatial data and cannot be eliminated.

Another explanation for the poorer MAE performance from SPIReS-MODIS is that some spatial variation in topography is lost with the coarser resolution. To test this hypothesis, a semi-variogram of the terrain slope is examined, as in Baba et al.

(2019). The semi-variogram shows a flattening around 500 m, indicating that variation in topography, which can manifest in topographically driven variables such as direct solar illumination, is poorly captured at MODIS and coarser spatial scales. This semi-variogram analysis confirms the above hypothesis. Further, downscaling coarse scale reanalysis products (Winstral et al., 2014) e.g., the downwelling radiation from Clouds and Earth's Radiant Energy System (Rutan et al., 2015) at 1° spatial resolution, has inherent limitations, often due to clouds (Lapo et al., 2017). Important to note is that ParBal does not use

precipitation as a forcing and thus does not suffer from well-known biases and downscaling issues (Raleigh et al., 2015; Pflug et al., 2021).

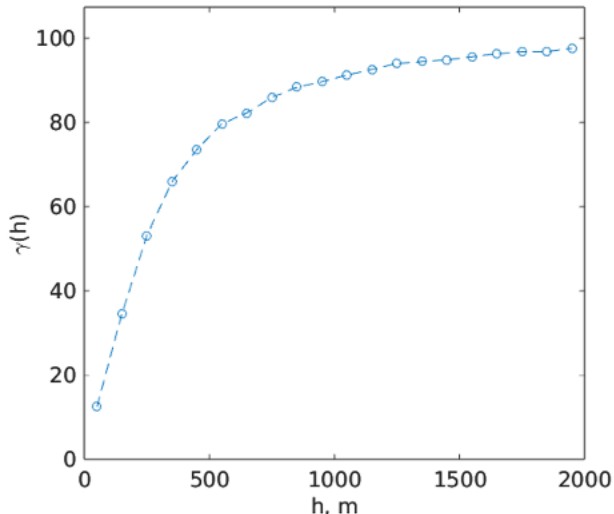

Figure 4:

*Semi-variance of terrain slope of the Tuolumne River Basin above Hetch Hetchy. The slope of the semivariance (not the terrain*
*slope itself) shows a flattening around 500 m, or about the MODIS pixel size.*

Alternatively, the lower MAE may indicate the importance of daily imaging from MODIS compared to the HLS snow cover, which had median gap of 5 days between revisits after filtering for clouds. In contrast, the SCAG-Fusion used daily MODIS snow cover in the prediction and training steps indicating it suffers from errors not related to revisit time.

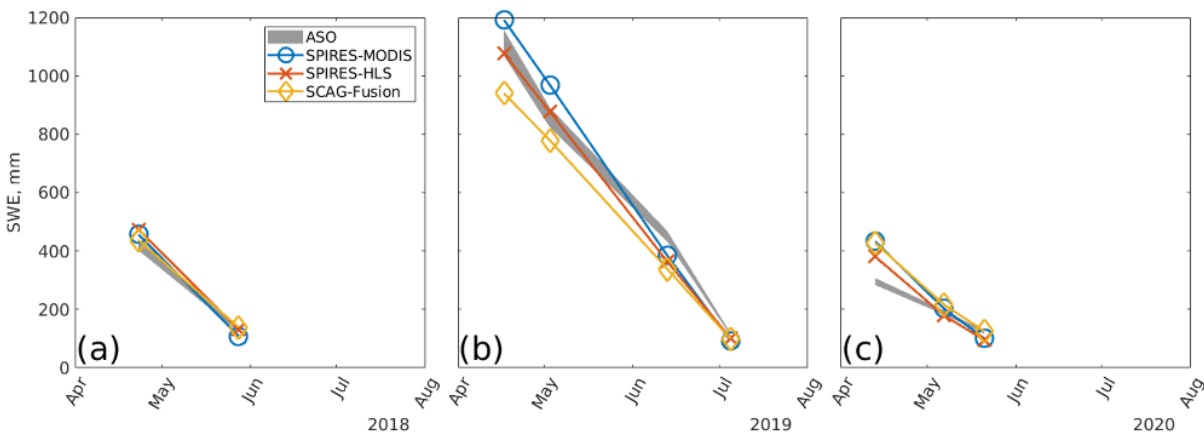

Figure 5:

*Basin-side SWE values by date for the three SWE reconstructions (a-c) compared to Airborne Snow Observatory estimates.*
*Assumed uncertainty in the ASO measurements is ±4% (Section 2.5) and is shaded in gray.*

| Name | Year | Bias, mm | Relative Bias, % | MAE, mm | Relative MAE, % |
|---|---|---|---|---|---|
| SPIReS-MODIS | 2018 | 9 | 3 | 87 | 32 |
| | 2019 | 26 | 4 | 168 | 26 |
| | 2020 | 43 | 21 | 95 | 47 |
| | **mean** | **26** | **9** | **117** | **35** |
| SPIReS-HLS | 2018 | 27 | 10 | 135 | 49 |
| | 2019 | -24 | -4 | 201 | 32 |
| | 2020 | 12 | 6 | 111 | 54 |
| | **mean** | **5** | **4** | **149** | **45** |
| SCAG-Fusion | 2018 | 13 | 5 | 122 | 44 |
| | 2019 | -89 | -14 | 264 | 42 |
| | 2020 | 50 | 24 | 123 | 60 |
| | **mean** | **-9** | **5** | **170** | **49** |

Table 3:

*Error statistics by year for the three SWE reconstructions. Mean values for all years are shown in **bold**. More detailed errors*
*by date are given in the Appendix.*

An example of the SWE modeled by ASO on 4 May 2019 and the three reconstructions is shown in Figure 6. The spatial distribution of the SWE from ASO matches well with all the reconstructions. Differences between the reconstructions can be seen around Mount Lyell, at the southernmost part of the basin. The ASO SWE shows high variability here, ranging from a few hundred mm of SWE to over 2000 mm, while the reconstructions model consistently higher amounts of SWE. The

overestimates here are likely related to false-positive classifications for snow. Especially late in the summer, when melt rates are high, these false-positives can lead to substantial overestimates of SWE during reconstruction (Slater et al., 2013). A close examination of the mostly snow-free areas in gray shows that only the SPIReS-HLS reconstructions replicate the small patches of thin snow in this area, likely because the SPIReS-HLS snow cover was not smoothed to the same degree as the SPIReS-MODIS or SCAG-Fusion, which both use heavy smoothing to reduce noise and smearing from MODIS.

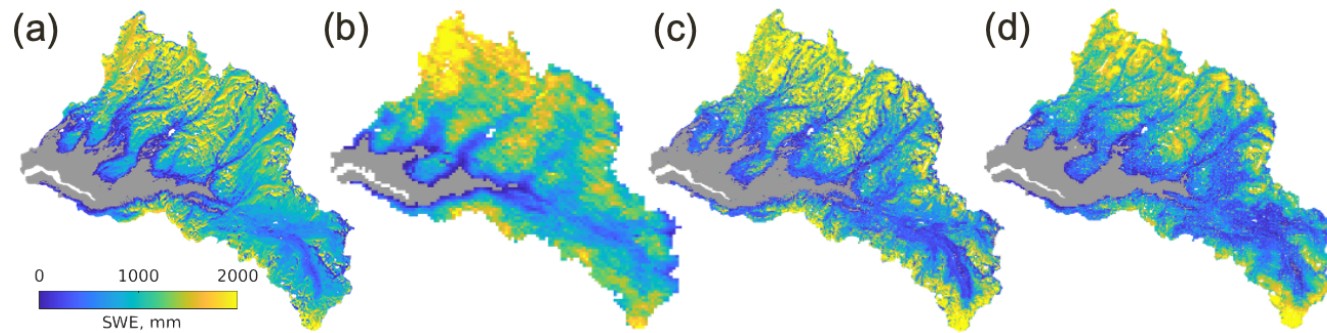


Figure 6:

*SWE in the Tuolumne River Basin above Hetch Hetchy Reservoir for 4 May 2019 modeled by the Airborne Snow Observatory (a) along with reconstructions from SPIReS-MODIS (b), SPIReS-HLS (c), and SCAG-Fusion (d).*

Errors are further examined by date (Figure 7 and Table A2). Except for 13 Apr 2020, the bias across all the products is
between -20 and 20% (Figure 7a). Figure 8 shows a snow pillow (weighing gauge, California Department of Water Resources station code DAN, elevation 2987 m) and that the ASO flight on 13 Apr 2020 is the only flight in this study that occurred prior to peak SWE. Overestimates of SWE prior to its peak are a limitation of SWE reconstruction. The hybrid SWE method (Section 2.4) extends SWE estimates throughout the year, but the high biases found on this date are not surprising, because snow melt occurred prior to the flight and snow accumulation occurred after the flight. Note the missing data on DAN after 3 May 2018
in Figure 8, but the CUES snow pillow, which is nearby and at a similar elevation (2940 m), shows clear ablation during May 2018.

Examination of the per-pixel MAE (Figure 7b) shows that the SPIReS-HLS product has the most consistent values, with the two approaches that used MODIS data (SPIReS-MODIS and SCAG-Fusion), showing more variability, perhaps again due to the smoothing needed for the relatively noisy MODIS data or the fact that SCAG-Fusion was trained using more data outside
the test period (March 2013 to December 2017 and January 2021 to March 2022) than within it (January 2018 to January 2021).

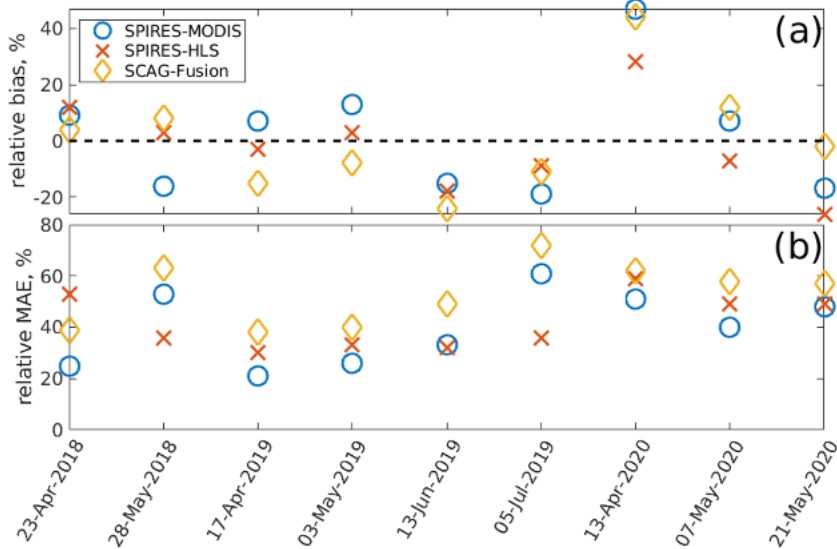

Figure 7:

*SWE errors by date for the three SWE reconstructions. The relative bias is shown in (a), the relative MAE in (b).*

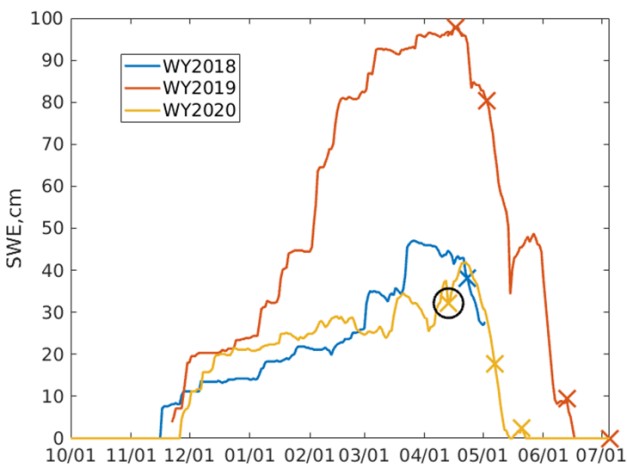


Figure 8:

*Snow pillow DAN in the Tuolumne River Basin showing daily SWE. The X markers show the dates of ASO flights. Circled is the 13 Apr 2020 ASO flight, which is the only flight that occurred prior to peak SWE. The pillow was not reporting from 3 May 2018 to 21 Nov 2018, but a nearby snow pillow shows consistent ablation in May 2018.*

Stillinger et al. (2023) show that errors in snow cover mapping depend on canopy cover, having to do with how much areal snow is viewable at the pixel scale by a sensor, which affects the accuracy of the SWE reconstructions (Bair et al., 2016). Thus, we examine errors in the SWE reconstructions, binned by canopy cover fraction, for each snow cover forcing. The bin centered at 5% (range: 0 to 9.9%) canopy cover (containing 46-60% of pixels in the basin, Table A3) shows (Figure 9ab)

relatively unbiased errors with MAE values close to the means (Table 3), but SWE biases become positive with increasing canopy cover for SPIReS-MODIS, yet negative for SCAG-Fusion and for SPIReS-HLS (except for the highest canopy fractions which contain only 5% of the basin's pixel, Table A3).

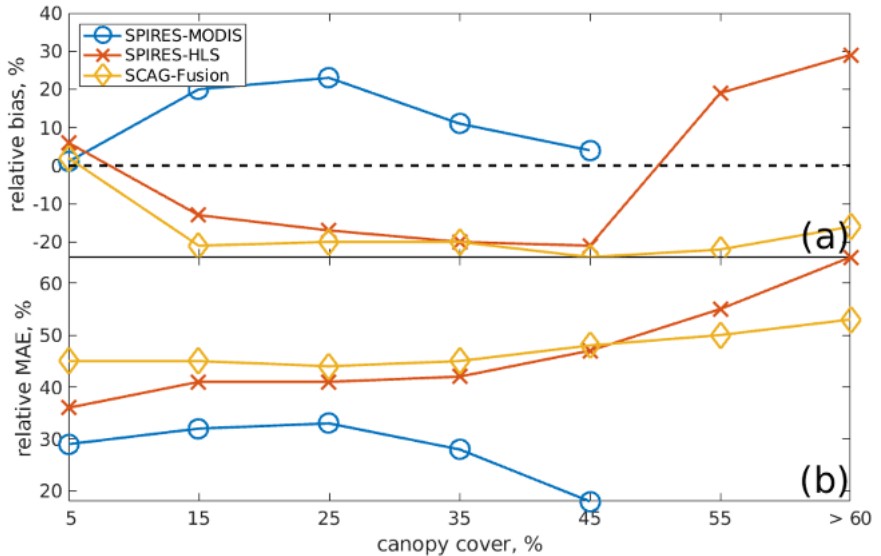

Figure 9:

*SWE errors for all dates for the three SWE reconstructions binned by canopy cover percent. Labeled are the bin centers. The relative bias is shown in (a), the relative MAE in (b).*

The bias and MAE with increasing canopy cover for SPIReS-HLS and SCAG-Fusion SWE reconstructions are similar to errors in fsca from Landsat 8 (Figure 4c, Stillinger et al., 2023). These fsca biases have similar shapes to the SWE biases indicating these fsca errors cause the SWE errors. Conversely, SPIReS-MODIS shows unbiased fsca with increasing canopy cover (Figure 5d, Stillinger et al., 2023), indicating some other source of error in the SPIReS-MODIS SWE reconstructions.

In summary, the answer to the question posed by the title of this study is that basin-wide SWE is marginally more accurate with finer spatial resolution. Specifically, the bias–arguably the most important error statistic for water resource management– was 4-5% lower using the finer resolution snow cover forcings. However, the results are mixed relative to previous studies. For example, Durand et al. (2008) and Molotch and Margulis (2008) report both lower MAE and bias with a 30 m Landsat ETM+ snow cover forcing compared to snow cover from MODIS and AVHRR. The explanation for why some previous studies showed more significant improvements going from moderate to high resolution forcings may be the snow mapping algorithms used. An accurate technique for dealing with mixed pixels is particularly important for moderate resolution sensors since in for mid-latitude mountains most pixels are mixed at 500 m (Selkowitz et al., 2014). In Durand et al. (2008) and Molotch and Margulis (2008), the finer resolution Landsat ETM+ snow cover used a spectral unmixing technique (Painter et al., 2003), but the MODIS snow cover was based on the Normalized Snow Difference Technique, which only uses two bands, versus all available for spectral unmixing, and is shown to have higher MAE and bias (Stillinger et al., 2023). In Cline et al.

(1998), the only other study to specifically examine spatial scale with SWE reconstruction, a spectral mixture technique was used on 30 m Landsat ETM+ to produce snow cover estimates (Rosenthal and Dozier, 1996). In that study, the coarsened results produced basin-wide SWE above and below the control simulation used as validation, suggesting that coarsening components of the energy balance did not show a clear trend in error. The snow cover used in that study is shown to have low bias and other measures of error from [0-1] fsca (Rosenthal and Dozier, 1996), thus reducing errors from mixed pixels. Increased spatial and temporal resolution through sensor design, fusion techniques, and satellite constellations are the future of Earth observations, but this study shows how a  moderate resolution sensor such as MODIS still offers value for snow mapping and modeling.

## 4. Conclusion

Optimal resolution questions are fundamental to the global study of snow and will inform future scientific priorities and mission specifications. Increasing spatial and temporal resolution mark remote sensing achievements with the implicit assumption that finer resolution provides greater accuracy. To test this assumption for snow hydrology, an energy balance SWE reconstruction model was run at two different spatial resolutions using three different snow cover forcings. Contrary to previous work, the baseline case using SPIReS-MODIS, a daily 463 m product, showed a lower MAE–a measure of per-pixel accuracy–compared to SCAG-Fusion and SPIReS-HLS, both with 30 m spatial resolution. The SPIReS-HLS showed the lowest bias, however the differences in the errors between all three products may be within the uncertainty caused by scaling artifacts such as basin boundary delineation. The improved bias with increasing spatial resolution, arguably the most important measure for water management, is a promising result; however the increased MAE with finer spatial resolution suggests that the daily acquisitions from MODIS with finer *temporal* resolution provide additional accuracy and/or that there are downscaling limitations with relatively coarse reanalysis data e.g., $10^5$ m (1º) downscaled to 30 m. Improvements such as the inclusion of Landsat 9 and version 2.0 of the HLS data may improve some of the errors. Future satellite missions that leverage existing and planned constellations such as Landsat Next will improve revisit times, as gaps between observations are still an issue for the HLS data. In summary, conclusions are: 1) Spectrally unmixed snow cover and snow albedo from MODIS continues to provide accurate forcings for snow models and 2) increased spatial and temporal resolution through sensor design, fusion techniques, and satellite constellations are the future of Earth observations, but existing moderate resolution sensors still offer value.

## Code availability

The codes for ParBal and SPIReS are available on GitHub: https://github.com/edwardbair.
The code for SCAG products is not available.

## Data availability

All data are in accessible repositories.

SPIReS-MODIS: The snow cover is part of a daily Western US product covering WY 2001-2021 (Bair and Stillinger, 2022).

The corresponding reconstructed SWE is in a Dryad Repository (Bair, 2023a).

SPIReS-HLS: The snow cover and reconstructions are in a Dryad repository (Bair, 2023b).

SCAG-Fusion: The snow cover and reconstructions are in a Dryad repository (Bair, 2023b).

**Author contribution**

According to CRediT taxonomy:

EHB - all 14 contributor roles

JD - Conceptualization, writing (review & editing)

KR - Conceptualization, data curation, formal analysis, funding acquisition, methodology, resources, writing (review & editing)

TS - Investigation, writing (review & editing)

WK - Data curation, formal analysis, investigation, methodology, software, supervision

RED - Resources, funding acquisition

**Competing interests**

The first author is a member of The Cryosphere Editorial Board.

**Acknowledgements**

This research was supported by NASA awards: 80NSSC21K0997, 80NSSC20K1722, 80NSSC20K1349, 80NSSC18K1489, 80NSSC21K0620, 80NSSC18K0427, 80NSSC20K1721, 80NSSC22K0703, & 80NSSC22K0929. Other support is from

Broad Agency Announcement Program and the Cold Regions Research and Engineering Laboratory (ERDC-CRREL) under Contract No. W913E521C0001. We thank Simon Gascoin and one anonymous referee for their insightful reviews.

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

**Appendix A**

| Name | Year | Bias, mm | Relative Bias, % | MAE, mm | Relative MAE, % |
|---|---|---|---|---|---|
| SPIReS-MODIS | 2018 | 9 | 3 | 87 | 32 |
| | 2019 | 26 | 4 | 168 | 26 |
| | 2020 | 43 | 21 | 95 | 47 |
| | **mean** | **26** | **9** | **117** | **35** |
| SPIReS-HLS | 2018 | 40 | 15 | 140 | 52 |
| | 2019 | -6 | -1 | 194 | 31 |
| | 2020 | 26 | 13 | 100 | 50 |
| | **mean** | **20** | **9** | **145** | **44** |
| SCAG-Fusion | 2018 | 1 | 0 | 90 | 33 |
| | 2019 | -108 | -17 | 221 | 35 |
| | 2020 | 52 | 26 | 97 | 48 |
| | **mean** | **-18** | **3** | **136** | **39** |

Table A1:

*Error statistics by date for the three SWE reconstructions, but with all pixels upscaled to 2000 m. The SPIRES-MODIS rows*
*are identical to those in Table 2 and are shown for comparison.*

| Name | Date | Bias, mm | Bias, % | MAE, mm | MAE, % |
|---|---|---|---|---|---|
| SPIReS-MODIS | 23 Apr 2018 | 38 | 9 | 107 | 25 |
| | 28 May 2018 | -20 | -16 | 68 | 53 |
| | 17 Apr 2019 | 79 | 7 | 228 | 21 |
| | 03 May 2019 | 111 | 13 | 223 | 26 |
| | 13 Jun 2019 | -66 | -15 | 150 | 33 |
| | 05 Jul 2019 | -22 | -19 | 70 | 61 |
| | 13 Apr 2020 | 137 | 47 | 150 | 51 |
| | 07 May 2020 | 13 | 7 | 76 | 40 |
| | 21 May 2020 | -21 | -17 | 59 | 48 |
| SPIReS-HLS | 23 Apr 2018 | 51 | 12 | 225 | 53 |
| | 28 May 2018 | 3 | 3 | 46 | 36 |
| | 17 Apr 2019 | -32 | -3 | 336 | 30 |
| | 03 May 2019 | 28 | 3 | 284 | 33 |
| | 13 Jun 2019 | -79 | -18 | 142 | 32 |
| | 05 Jul 2019 | -10 | -9 | 41 | 36 |
| | 13 Apr 2020 | 84 | 28 | 175 | 59 |
| | 07 May 2020 | -14 | -7 | 95 | 49 |
| | 21 May 2020 | -33 | -26 | 63 | 49 |
| SCAG-Fusion | 23 Apr 2018 | 15 | 4 | 164 | 39 |
| | 28 May 2018 | 10 | 8 | 80 | 63 |
| | 17 Apr 2019 | -169 | -15 | 419 | 38 |
| | 03 May 2019 | -70 | -8 | 338 | 40 |
| | 13 Jun 2019 | -105 | -24 | 216 | 49 |
| | 05 Jul 2019 | -13 | -11 | 81 | 72 |
| | 13 Apr 2020 | 131 | 44 | 185 | 62 |
| | 07 May 2020 | 23 | 12 | 111 | 58 |
| | 21 May 2020 | -3 | -2 | 74 | 57 |

Table A2:

*Error statistics by date for the three SWE reconstructions.*

| Name | Canopy cover, % | Pixels, number | Pixels, % | Bias, mm | Bias, % | MAE, mm | MAE, % |
|---|---|---|---|---|---|---|---|
| SPIReS-MODIS | 5 | 131 | 46 | 7 | 1 | 155 | 29 |
| | 15 | 76 | 27 | 68 | 20 | 111 | 32 |
| | 25 | 38 | 13 | 57 | 23 | 81 | 33 |
| | 35 | 25 | 9 | 23 | 11 | 62 | 28 |
| | 45 | 8 | 3 | 8 | 4 | 36 | 18 |
| | 55 | 0 | 0 | | | | |
| | > 60 | 0 | 0 | | | | |
| SPIReS-HLS | 5 | 48230 | 60 | 27 | 6 | 167 | 36 |
| | 15 | 8737 | 11 | -45 | -13 | 143 | 41 |
| | 25 | 6553 | 8 | -59 | -17 | 137 | 41 |
| | 35 | 6969 | 9 | -64 | -20 | 132 | 42 |
| | 45 | 5760 | 7 | -59 | -21 | 133 | 47 |
| | 55 | 2869 | 4 | 52 | 19 | 152 | 55 |
| | > 60 | 962 | 1 | 73 | 29 | 165 | 65 |
| SCAG-Fusion | 5 | 48230 | 60 | 11 | 2 | 212 | 45 |
| | 15 | 8737 | 11 | -74 | -21 | 156 | 45 |
| | 25 | 6553 | 8 | -67 | -20 | 148 | 44 |
| | 35 | 6969 | 9 | -63 | -20 | 141 | 45 |
| | 45 | 5760 | 7 | -70 | -24 | 137 | 48 |
| | 55 | 2869 | 4 | -60 | -22 | 136 | 50 |
| | > 60 | 962 | 1 | -41 | -16 | 135 | 53 |

Table A3:

*Error statistics by canopy cover, for all dates, for the three SWE reconstructions.*