# Peer review of "How do tradeoffs in satellite spatial and temporal resolution impact snow water equivalent reconstruction?"

_The Cryosphere, 2022_

## Author Comment (AC1)

Dear Reviewer #1:

Thank you for reviewing our manuscript. I appreciate your time and critiques.

Your comments are in blue, our responses are in black, and text to be included in our revision is in red.

This article has an attractive title which unfortunately does not reflect its content. The authors present a comparison of the results obtained by backward reconstruction of the SWE from three different remote sensing products. The rationale for this study is actually more specifically given at the end of the introduction and is "considerable advances have been made in SWE reconstruction techniques as well as snow cover and albedo mapping, hence the justification for revisiting the effects of spatial and temporal resolution". But the advances in question are not specified. And the question of the albedo is not really studied in the rest of the paper.

Thank you for pointing this out. Our intention was to show that the advances in snow cover & albedo mapping as well as snow water equivalent reconstruction have been well-published, with many of the publications by the authors of this manuscript. We've inserted references into the quoted sentence.

Since those studies, considerable advances have been made in SWE reconstruction techniques (Bair et al., 2016; Rittger et al., 2016) as well as snow cover (Stillinger et al., 2023) and albedo mapping (Bair et al., 2019), hence the justification for revisiting the effects of spatial and temporal resolution.

Significant work has been done to perform these simulations, but the analysis of the results remains superficial and does not explore the mechanisms that explain the effects of the resolution on the modeling of snow cover.

We agree that the mechanisms explaining the effects of resolution are not fully explored. More on this below.

However, the conclusion that seems to emerge is that the resolution has no impact on the estimate of the resource, which is counterintuitive when compared to studies that immediately come to mind because I contributed to them (Baba et al. 2020, Bouamri et al. 2021) or earlier by Schlögl et al. (2016).

The characterization "that resolution has no impact on the estimate of the resource" is not accurate. The findings are that the basin-wide error, measured by bias, decreases with increasing resolution, while the per-pixel accuracy, measured by MAE, increases.

Likewise, previous studies show mixed results when increasing spatial resolution of the snow models. These studies are detailed in the literature review in the Introduction. We've added a reference to Schlögl et al. (2016) in the chronological literature review.

Contrary to Cline et al. (1998), Schlögl et al. (2016) report that SWE increases with DEM resolution in two alpine basins.

Baba et al. (2019) was also recommended by the Associate Editor, so we've included that as well.

Similarly, Baba et al. (2019) used an energy balance model with a DEM at 8-1000 m and report good agreement with fine resolution snow cover maps up to 250 m, but a loss in agreement at coarser resolution, likely due to excessive smoothing of topographic effects.

Bouamri et al. (2021) is also now in the Introduction.

Bouamri et al. (2021) examined differences between snowmelt models with and without solar radiation represented. Although the models with solar radiation better simulated the snowcover used for validation, aggregating the simulated snow cover from 100 to 500 m suppressed those improvements.

The fact that the source products also have different revisit times (and different processing algorithms) complicates the analysis of the effect of spatial resolution.

This is a good point. However, temporal vs spatial resolution is a fundamental tradeoff in remote sensing. As we point out in the Introduction, there is no single sensor that offers high spatial and temporal resolution. Further, Stillinger et al. (2023) show minor differences in snow mapping performance between the SCAG and SPIReS.

In fact, the discussion concerns the artefacts linked to the delimitation of the watershed, which does not seem to me to be a central issue.

We disagree. Comparing the same area across datasets is essential in spatial scale studies. We note that delineation artifacts were also reported by Cline et al. (1998) who normalized by the high resolution basin area, which was smaller than coarser spatial resolution model runs. However in the study, the opposite occurs, the basin decreases in size with resolution, illustrating differences in basin outlines and aggregation techniques. Starting with the third sentence of Section 3, we explain in detail how we arrived at the conclusion that these artifacts cannot be eliminated.

I think it is necessary to help the reader to interpret the results, perhaps through an analysis of the energy balance or semi-variograms of topographic variables.

Thank you for this suggestion. An energy balance analysis is problematic, as there is no validation source. We know qualitatively that variables like direct solar radiation should resemble the topography, but without a reliable spatially-distributed validation source, one cannot say whether, for example, the basin-wide mean direct solar radiation is more accurate for the fine or coarse spatial resolution product.

A variogram analysis provides an estimate of the scale break, thus as in Baba et al. (2019), we've added a semi-variogram analysis. Similarly, we find that the slope of the variogram flattens around 500 m, suggesting that the slope angle variation is not sufficiently captured by MODIS in the Tuolumne River Basin above Hetch Hetchy.

Another explanation for the poorer MAE performance from SPIReS-MODIS is that some spatial variation in topography is lost with the coarser resolution. To test this hypothesis, a semi-variogram of the terrain slope is examined, as in Baba et al. (2019). The semi-variogram shows a flattening around 500 m, indicating that variation in topography, which can manifest in topographically driven variables such as direct solar illumination, is poorly captured at MODIS and coarser spatial scales. This semi-variogram analysis confirms the above hypothesis.

[Figure]

Figure 1:
*Semi-variance of terrain slope of the Tuolumne River Basin above Hetch Hetchy. The slope of the semivariance (not the terrain slope itself) shows a flattening around 500 m, or about the MODIS pixel size.*

Another thing puzzled me when reading the manuscript. The authors introduce a method for reconstructing the SWE before the accumulation peak which is a rescaling of the GLDAS SWE. This SWE is therefore produced from different forcings, which further complicates the interpretation of the results.

We agree this is confusing. A major limitation of SWE reconstruction is that it provides no information prior to peak SWE and further, without ancillary data such as snow pillow measurements (Rittger et al., 2016), the peak date is unknown. Scaling the GLDAS SWE prior to the peak is a method to overcome this limitation. As Figure 7 shows, 7/8 ASO flights over the Tuolumne occurred at or post peak SWE, thus the rescaling method, which we admit has not been validated during the accumulation season, has little impact on the results of this study. In any case, the same reconstruction methodology was applied to each of the three snow cover forcings, so any limitations or advantages related to modeling dates prior to peak SWE are consistent across the study.

Equation (1) is incomprehensible to me*.

Agree, thank you for highlighting this mistake. Equation 1 has been changed to

Pixels with the same snow cover duration are identified as

$$SWE^*_{GLDAS} = \left(SWE_{GLDAS,\Delta t_1} > 0, fsca_{\Delta t_1} > 0\right) \& \left(SWE_{GLDAS,\Delta t_2} = 0, fsca_{\Delta t_2} = 0\right) \qquad (1)$$

where the asterisk denotes the selected pixels and fsca is from the fine-scaled product e.g., Section 2.1 - 2.2. The $\Delta t_1$ and $\Delta t_2$ indicate different time periods.

The final conclusion of the article "increased spatial and temporal resolution (...) are the future of Earth observations." could have been written before carrying out this study and concerns many other fields of application than snow. However, it does not seem to me that the results and the very design of the study support this conclusion.

Changed to the text below. Also made a similar change to the end of the Abstract.

In summary, conclusions are: 1) Spectrally unmixed snow cover and snow albedo from MODIS continues to provide accurate forcings for snow models and 2) increased spatial and temporal resolution through sensor design, fusion techniques, and satellite constellations are the future of Earth observations, but existing moderate resolution sensors still offer value.

In terms of presentation, the authors introduce additional analyzes in the results section which have not been presented in the method section as recommended for scientific articles.

Without an example, it's difficult to address this point. Perhaps the Reviewer is referring to the snow pillow data in Figure 7? Note the section is titled "Results and discussion." Presenting ancillary data, such as snow pillow data, or additional analysis is common in discussion sections. For example, in Baba et al. (2019), the semivariogram analysis is presented without any previous mention in the methods section for how the analysis was conducted.

In the end, all this leads me to think that this manuscript was prepared a little too quickly, which is regrettable given the work and calculation behind the production of these datasets. I'm sorry to give such a negative review, maybe another reviewer will disagree with me.

* The $\wedge$ operator is an "n-ary logical and" so the result should not be a SWE value but a boolean (vector) variable. Besides I don't understand if the pixels are selected by considering the time series of SWE and fsca (the time index does not appear).

See the revised Equation above

NB) I was unable to get the data from the ftp server indicated at the end of the manuscript. The connection is possible but not the download (I tried from two different networks)

Thank you for testing. The FTP server was having issues that have been fixed.

L66: found

Present tense (find) is used correctly here as it is used widely in scientific writing when describing results. "Winstral et al...find..." NOT "Winstral et al...found."

L89: the bowtie effect of MODIS acquisitions was known before this reference

Added Wolfe et al. (1998)

The baseline case uses MODIS at 463 m daily resolution, although the effective pixel size can be up to 5× as large for off nadir acquisitions (Wolfe et al., 1998; Dozier et al., 2008)

L98: parenthesis

Fixed

L98: any reason why HLS v2 was not available? What is the difference with v1 and would it change the results?

Yes, as explained on that line the data are incomplete in the NASA Earthdata Search archive for HLS v 2.0

L105: any clue why the revisit is not 2-3 days?

2-3 days is an unrealistic specification globally. The User's Guide confirms that the theoretical mean revisit of 3.8 days is close to what we found, 3.5 days.

L114: why eliminate certain images after visual inspection? this seems incompatible with a global application ("global snow").

As stated, the issue is that many of the scenes were 100% cloud covered or missing most of the watershed. This imaging pruning could be automated.

L124: This should be explained ("a second cloud filtering step using Superpixels and Gabor filtering was used")

Reference added

Because the initial filtering did not remove all clouds, a second cloud filtering step with Superpixels and Gabor filtering was used (Stillinger, 2019)

L131: "SPIReS, SCAG, and all other accessible snow mapping algorithms" I have checked this article and this assertion is incorrect.

changed to

For more details and a recent comparison between SPIReS, SCAG, and other snow mapping algorithms see Stillinger et al. (2022)

L173: why not use all ASO acquisitions? There are many more on this basin since 2017.

We were limited by the SCAG-fusion output, which did not extend past 2020.

L184: can you specify or indicate the tool? "using a mean-preserving technique with a weighted resampling covering the image". Imagine that a reader would like to use the same approach (I would).

Added a citation.

The ASO images were resampled from a cell size of 50 m to 2000 m (4× the MODIS resolution) and 120 m (4× the Landsat resolution), using a mean-preserving technique with a weighted resampling covering the image (mapresize, MathWorks, 2022).

L185: the geolocation accuracy of S2 is about 1 pixel of 10m, not 1-2 pixels of 30m. See the data quality reports by ESA. Note that recent GRI reprocessing should result to subpixel accuracy (<10m). Also, Storey et al. (2016) report that Landsat OLI has a geolocation accuracy of 18 meters (CE90), not 1-2 pixels of 30m.

Storey et al. (2016) write "...however, the Landsat-8 framework, based upon the Global Land Survey images, contains residual geolocation errors leading to an expected sensor-to-sensor misregistration of 38 m (2σ)." To our knowledge, this sensor-to-sensor error has yet to be fixed, as acknowledged in Claverie et al. (2018), the article describing the HLS dataset.

Thus, it is not unreasonable to assume a geolocational uncertainty of 1-2 pixels when comparing products from the two sensors.

Changed to

The upscaled cell sizes account for geolocational and sensor-to-sensor uncertainty of 1-2 pixels for MODIS and Landsat/Sentinel-2 (Tan et al., 2006; Storey et al., 2016).

L204: I may have missed somethin but why not make this comparison for other products? as it stands, this part on the albedo does not add much to the study.

The comparison has already been done for the SCAG product, as stated in the next two sentences. "These improvements in remotely sensed snow albedo over previous assessments, showing RMSE values of 4.6 to 4.8% with 0.7-1.3% bias for MODIS (Bair et al., 2019; Bair et al., 2021)." The HLS product albedo has yet to be evaluated for snow albedo accuracy. It's likely that the errors are similar, given similar surface reflectance errors to MODIS.

The point is that the snow albedo errors are comparable to the surface reflectance errors and are quite low, thus they are not a major source of uncertainty.

L271: shown

Fixed

L305: S2C should replace S2A hence it will not improve revisit time (except for a short period). https://labo.obs-mip.fr/multitemp/some-news-from-esa-regarding-the-coming-sentinels-1-and-2/

Changed to

[revised manuscript text omitted]

---

## Author Comment (AC2)

Dear Reviewer #2:

Thank you for reviewing our manuscript. I appreciate your time and critiques.

Your comments are in blue, our responses are in black, and text to be included in our revision is in red.

In this paper, the authors examine the value of two recently developed satellite fusion products, combined with reconstruction models to produce high spatial and temporal resolution SWE data. Snow covered area (SCA) data produced through the fusion of MODIS and Landsat or Sentinel and Landsat have the potential to provide high spatial and temporal resolution data that is not available through any single sensor. This study evaluates these products along with a baseline MODIS-derived SCA product to assess the effects of spatial and temporal resolution on SWE estimates. The results found that while the bias is lower for the high resolution products, the mean absolute error is higher which is different than previous studies which found better results with higher resolution data.

This work is relevant and timely to the snow community and to ongoing discussions about the measurement requirements of future satellite missions. It contributes to recent work on fusing various data products together for improved spatial and/or temporal resolution snow observations. The manuscript is well written, and I believe it will be ready for publication with minor revision. However, there are a few areas where I think the manuscript could be improved.

The authors stop short of answering the compelling question posed by the title, or even going into much discussion on it. The results seem to suggest the answer is no, but one of the primary conclusions is that we're headed that way (towards high resolution data) anyway. If that title is kept then I think the discussion needs to be greatly expanded to cover why these results may differ from previous studies.

Thank you for the suggestion. We have added the and editing the following in the Results and Discussion

In summary, the answer to the question posed by the title of this study is yes, as the bias– arguably the most important error statistic for water resource management–was 4-5% lower using the higher resolution snow cover forcings. However, the results are mixed relative to previous studies. For example, Durand et al. (2008) and Molotch and Margulis (2008) report both lower MAE and bias with a 30 m Landsat ETM+ snow cover forcing compared to snow cover from MODIS and AVHRR. The explanation for why some previous studies showed more significant improvements going from moderate to high resolution forcings may be the snow mapping algorithms used. An accurate technique for dealing with mixed pixels is particularly important for moderate resolution sensors since in for mid-latitude mountains most pixels are mixed at 500 m (Selkowitz et al., 2014). In Durand et al. (2008) and Molotch and Margulis (2008), the higher resolution Landsat ETM+ snow cover used a spectral unmixing technique (Painter et al., 2003), but the MODIS snow cover was based on the Normalized Snow Difference Technique, which only uses two bands, versus all available for spectral unmixing, and is shown to have higher MAE and bias (Stillinger et al., 2023). In Cline et al. (1998), the only other study

to specifically examine spatial scale with SWE reconstruction, a spectral mixture technique was used on 30 m Landsat ETM+ to produce snow cover estimates (Rosenthal and Dozier, 1996). In that study, the coarsened results produced basin-wide SWE above and below the control simulation used as validation, suggesting that coarsening components of the energy balance did not show a clear trend in error. The snow cover used in that study is shown to have low bias and other measures of error from [0-1] fsca (Rosenthal and Dozier, 1996), thus reducing errors from mixed pixels. Increased spatial and temporal resolution through sensor design, fusion techniques, and satellite constellations are the future of Earth observations, but this study shows how a moderate resolution sensor such as MODIS still offers value for snow mapping and modeling.

This manuscript could also help initiate a discussion on the value of high resolution data and what is required to outweigh the cost associated with increased data storage and processing time, particularly at a global scale. Based on the results of this analysis, is it worth it? If not, what improvement would be needed (i.e. error reduced by how much) to make it worth it? Alternatively, you could change the title to reflect the current content, e.g. analysis of recent snow cover data fusion products to drive SWE reconstruction models.

Questions of economic value, which is what I assume the Reviewer is referencing, are interesting and worthwhile, but beyond the scope of this study. One issue is that the costs and benefits are difficult to quantify. For example, the price of water in California fluctuates, based on who is purchasing it, how much they are purchasing. The economics of water is a well-studied field which would be suitable to answer these questions. Similarly, satellite missions greatly differ in cost. For example, strategies to lower costs such as building multiple instruments at one time can lower the cost significantly. Here we answer the question posed by the title with error metrics, and as the above paragraph shows, the answer is yes.

Additional comments:

Line 50-51: The sentence "When these artifacts were corrected, the SWE volumes at 90m were overestimates and underestimates at coarser resolutions" is worded awkwardly. Suggest rewording to make it clearer.

Changed to

When these artifacts were corrected, the SWE volumes at 90 m were overestimates while those at coarser resolutions were underestimates.

Line 51: "showed" instead of "show"

This is confusing. The word "shows" not "show" appears on l 51 and is used correctly. Present tense (show) is used correctly here as it is used widely in scientific writing when describing results. "Blöschl...shows..." NOT Blöschl...showed..."

Line 65: "false negatives" – if MODIS has less patchy snow, I assume that means it was mapping full coverage (overestimating)? Should that say "fewer false positives" instead of "false negatives", like on line 248?

No, MODIS tends to miss snow and produce false negatives.

In Figure 6 of Rittger et al. (2013), relative to SCAG applied to ETM+, MODSCAG underestimates snow cover in the Himalaya and Upper Rio Grande, but slightly overestimates in the Sierra Nevada. For the two cases with underestimates, there are more pixels mapped as snow by ETM+, suggesting higher Recall ( TP / (TP + FN) ) where TP is true positive and FN is false negative. In other words, the issue is that MODSCAG is showing no snow on patchy pixels mapped with snow by ETM+, meaning MODSCAG has a higher proportion of false negatives than SCAG applied to ETM+. This makes intuitive sense as there's a lower limit to fsca detection. For example, say an isolated 10% snow covered pixel at 30 m with no other snow covered neighbors can be correctly mapped as snow by ETM+ whereas that pixel will be mapped as no snow when the same algorithm is applied to MODIS.

Line 66-67: Did Winstral et al. find that 100m resolution was needed for the forcing data, or the model resolution?

They tested both. Changed to

Winstral et al. (2014) examined scale in a snow energy balance model at a range of spatial resolutions and find that 100 m spatial resolution is needed to accurately simulate snow melt.

Line 120: what is "CFmask"? Not defined in text

Changed to

Standard cloud masks (Foga et al., 2017) were used to select the 100 most cloud free Level 2 surface reflectance images (USGS, 2021) for dates spanning Mar 2013 to Mar 2021 (**Error! Reference source not found.**).

Lines 154-169: This section is difficult to follow.  It sounds like you're using a domain average peak SWE and date to correct GLDAS. Why not correct it by pixel? A graph showing the basin-average SWE with the original GLDAS, ParBal, Hybrid SWE and ASO might help demonstrate the process.

Agree. Equation 1 has been revised to include time period subscripts. The correction is done by pixel. Sometimes there are multiple GLDAS pixels with matching snow duration to our remotely-sensed retrievals. In that case, an average between the GLDAS pixels is used.

Changed to

The concept is to identify GLDAS pixels with similar snow cover duration as the fine-scale fsca pixels, find the peak SWE day from those GLDAS pixels, then scale the GLDAS estimates by the ParBal SWE estimate on that peak day. This process is repeated for every fine-scale pixel.

Thank you for the suggestion to add a figure. We've added a figure showing the SWE for an individual pixel.

Figure 1 shows this hybrid GLDAS and reconstructed SWE for an example pixel in WY 2019. ParBal was run with each of the snow cover forcings, holding all other inputs constant.

[Figure]

Figure 1:
*Hybrid SWE estimates for the accumulation season combining reconstruction and GLDAS for an example pixel using the SPIReS-HLS snow cover.*

Line 232: The limitations of downscaling coarse resolution forcing data deserve more discussion, and additional references of more recent work (e.g. Pflug et al, 2021). "CERES" is mentioned for the first time here and not defined. While ParBal has been extensively covered in other papers, it seems worth describing the reanalysis datasets and downscaling techniques used in this study to better understand how that might impact the MAE.

Thank you for this reference. This section has been revised and now includes a semivariogram analysis per Reviewer #1's recommendation.

Another explanation for the poorer MAE performance from SPIReS-MODIS is that some spatial variation in topography is lost with the coarser resolution. To test this hypothesis, a semi-variogram of the terrain slope is examined, as in Baba et al. (2019). The semi-variogram shows a flattening around 500 m, indicating that variation in topography, which can manifest in topographically-driven variables such as direct solar illumination, is poorly captured at MODIS and coarser spatial scales. This semi-variogram analysis confirms the above hypothesis. Further, downscaling coarse scale reanalysis products (Winstral et al., 2014) e.g., the downwelling radiation from Clouds and Earth's Radiant Energy System (Rutan et al., 2015) at 1º spatial resolution, has inherent limitations, often due to clouds (Lapo et al., 2017). Important to note is that ParBal does use precipitation as a forcing and thus does not suffer from well-known biases and downscaling issues (Raleigh et al., 2015; Pflug et al., 2021).

[Figure]

Figure 2:
*Semi-variance of terrain slope of the Tuolumne River Basin above Hetch Hetchy. The slope of the semivariance (not the terrain slope itself) shows a flattening around 500 m, or about the MODIS pixel size.*

Lines 293 – 308, Conclusion: You state that the results differ from previous work, without going into a lot of detail. Specifically, how do the percent errors reported in Molotch and Margulis (2008) compare to the results of this study? They found almost the opposite MAE results using high-res and moderate-res data (lines 61-62). I think it would strengthen the paper to add more discussion on what is causing the differences in results.

We attribute it to a lack of spectral mixture analysis used for the MODIS snow retrievals. Please see the added text at the end of the Results and Discussion section referenced.

References:

Pflug, J. M., Hughes, M., & Lundquist, J. D. (2021). Downscaling snow deposition using historic snow depth patterns: Diagnosing limitations from snowfall biases, winter snow losses, and interannual snow pattern repeatability. *Water Resources Research*, **57**, e2021WR029999. https://doi.org/10.1029/2021WR029999

Baba, M. W., Gascoin, S., Kinnard, C., Marchane, A., and Hanich, L.: Effect of digital elevation model resolution on the simulation of the snow cover evolution in the High Atlas, Water Resources Research, 55, 5360-5378, 10.1029/2018WR023789, 2019.

Cline, D., Elder, K., and Bales, R.: Scale effects in a distributed snow water equivalence and snowmelt model for mountain basins, Hydrological Processes, 12, 1527-1536, 10.1002/(SICI)1099-1085(199808/09)12:10/11<1527::AID-HYP678>3.0.CO;2-E, 1998.

Durand, M., Molotch, N. P., and Margulis, S. A.: Merging complementary remote sensing datasets in the context of snow water equivalent reconstruction, Remote Sens Environ, 112, 1212-1225, 10.1016/j.rse.2007.08.010, 2008.

Foga, S., Scaramuzza, P. L., Guo, S., Zhu, Z., Dilley, R. D., Beckmann, T., Schmidt, G. L., Dwyer, J. L., Joseph Hughes, M., and Laue, B.: Cloud detection algorithm comparison and validation for operational Landsat data products, Remote Sens Environ, 194, 379-390, 10.1016/j.rse.2017.03.026, 2017.

Lapo, K. E., Hinkelman, L. M., Sumargo, E., Hughes, M., and Lundquist, J. D.: A critical evaluation of modeled solar irradiance over California for hydrologic and land surface modeling, Journal of Geophysical Research: Atmospheres, 122, 299-317, 10.1002/2016JD025527, 2017.

Molotch, N. P., and Margulis, S. A.: Estimating the distribution of snow water equivalent using remotely sensed snow cover data and a spatially distributed snowmelt model: A multi-resolution, multi-sensor comparison, Advances in Water Resources, 31, 1503-1514, 10.1016/j.advwatres.2008.07.017, 2008.

Painter, T. H., Dozier, J., Roberts, D. A., Davis, R. E., and Green, R. O.: Retrieval of subpixel snow-covered area and grain size from imaging spectrometer data, Remote Sens Environ, 85, 64-77, 10.1016/S0034-4257(02)00187-6, 2003.

Pflug, J. M., Hughes, M., and Lundquist, J. D.: Downscaling snow deposition using historic snow depth patterns: diagnosing limitations from snowfall biases, winter snow losses, and interannual snow pattern repeatability, Water Resources Research, 57, e2021WR029999, 10.1029/2021WR029999, 2021.

Raleigh, M. S., Lundquist, J. D., and Clark, M. P.: Exploring the impact of forcing error characteristics on physically based snow simulations within a global sensitivity analysis framework, Hydrol. Earth Syst. Sci., 19, 3153-3179, 10.5194/hess-19-3153-2015, 2015.

Rittger, K., Painter, T. H., and Dozier, J.: Assessment of methods for mapping snow cover from MODIS, Advances in Water Resources, 51, 367-380, 10.1016/j.advwatres.2012.03.002, 2013.

Rosenthal, W., and Dozier, J.: Automated mapping of montane snow cover at subpixel resolution from the Landsat Thematic Mapper, Water Resources Research, 32, 115-130, 10.1029/95WR02718, 1996.

Rutan, D. A., Kato, S., Doelling, D. R., Rose, F. G., Nguyen, L. T., Caldwell, T. E., and Loeb, N. G.: CERES synoptic product: Methodology and validation of surface radiant flux, Journal of Atmospheric and Oceanic Technology, 32, 1121-1143, 10.1175/JTECH-D-14-00165.1, 2015.

Selkowitz, D. J., Forster, R. R., and Caldwell, M. K.: Prevalence of pure versus mixed snow cover pixels across spatial resolutions in alpine environments, Remote Sensing, 6, 12478-12508, 10.3390/rs61212478, 2014.

Stillinger, T., Rittger, K., Raleigh, M. S., Michell, A., Davis, R. E., and Bair, E. H.: Landsat, MODIS, and VIIRS snow cover mapping algorithm performance as validated by airborne lidar datasets, The Cryosphere, 17, 567-590, 10.5194/tc-17-567-2023, 2023.

Winstral, A., Marks, D., and Gurney, R.: Assessing the sensitivities of a distributed snow model to forcing data resolution, Journal of Hydrometeorology, 15, 1366-1383, 10.1175/jhm-d-13-0169.1, 2014.

---

## Author Response (AR2)

Response to Associate Editor from 6/9/23

Dear Dr. Dumont:

Thank you for editing our manuscript. I appreciate your time and critiques.

Your comments are in blue, our responses are in black, and text to be included in our revision is in red.

Dear Authors,

Once again, thanks a lot for accounting for the comments from the editor and the referees and for changing the title.
I feel that the paper is now ready for publication.
There is one technical correction which is required by the referee :
"The authors have satisfactorily addressed my concerns. The new title reflects the content of the study. Only Equation 1 remains inconsistent since the result of the & operator should be a boolean, not a SWE value.

We have adjusted equation 1 as

Pixels with the same snow cover duration are identified by the logical vector $t$ as

$$t = \left(\text{SWE}_{GLDAS,\Delta t_1} > 0, \text{fsca}_{\Delta t_1} > 0\right) \& \left(\text{SWE}_{GLDAS,\Delta t_2} = 0, \text{fsca}_{\Delta t_2} = 0\right)$$
$$\text{SWE}^*{}_{GLDAS} = \text{SWE}_{GLDAS,t}$$

(1)

In addition it is not clear how \Delta t_1 and \Delta t_2 are defined.

We have added (days) on l 166

"...indicate different time periods (days)."

Note that the $\wedge$ symbol (instead of &) is used below for the same operation."
Can you take this into account ?

The $\wedge$ has been changed to an & in Eq 3

Other changes:

The data availability section has been updated , as the FTP links have been replaced with repositories with DOIs.

References were corrected to add DOIs and fix other errors

L 320 was changed to match the new title which is no longer a yes/no question.

In summary, the answer to the question posed by the title of this study is that basin-wide SWE is marginally more accurate with finer spatial resolution.